# Comparative Pathogenicity of *Aeromonas* spp. in Cultured Red Hybrid Tilapia (*Oreochromis niloticus × O. mossambicus*)

**DOI:** 10.3390/biology10111192

**Published:** 2021-11-17

**Authors:** Mohamad Azzam-Sayuti, Md Yasin Ina-Salwany, Mohd Zamri-Saad, Salleh Annas, Mohd Termizi Yusof, Md Shirajum Monir, Aslah Mohamad, Mohd Hafiz Ngoo Muhamad-Sofie, Jing Yie Lee, Yong Kit Chin, Zahaludin Amir-Danial, Addenan Asyiqin, Basri Lukman, Mark R. Liles, Mohammad Noor Azmai Amal

**Affiliations:** 1Aquatic Animal Health and Therapeutics Laboratory, Institute of Bioscience, Universiti Putra Malaysia, UPM Serdang 43400, Selangor, Malaysia; azzamsayuti96@gmail.com (M.A.-S.); mzamri@upm.edu.my (M.Z.-S.); annas@upm.edu.my (S.A.); monir_bau22@yahoo.com (M.S.M.); aslahumt@gmail.com (A.M.); sofiehafizngoo@gmail.com (M.H.N.M.-S.); leejingyie@gmail.com (J.Y.L.); chinyongkit@gmail.com (Y.K.C.); amirdanial6797@gmail.com (Z.A.-D.); asyiqinaddenan@gmail.com (A.A.); lukmanbasri95@gmail.com (B.L.); mnamal@upm.edu.my (M.N.A.A.); 2Department of Aquaculture, Faculty of Agriculture, Universiti Putra Malaysia, UPM Serdang 43400, Selangor, Malaysia; 3Department of Veterinary Pathology and Microbiology, Faculty of Veterinary Medicine, Universiti Putra Malaysia, UPM Serdang 43400, Selangor, Malaysia; 4Department of Microbiology, Faculty of Biotechnology and Biomolecular Sciences, Universiti Putra Malaysia, UPM Serdang 43400, Selangor, Malaysia; mohdtermizi@upm.edu.my; 5Department of Biological Sciences, Auburn University, Auburn, AL 36849, USA; lilesma@auburn.edu; 6Department of Biology, Faculty of Science, Universiti Putra Malaysia, UPM Serdang 43400, Selangor, Malaysia

**Keywords:** *myo*-inositol, LD_50_, *Aeromonas*, red hybrid tilapia, pathogenicity

## Abstract

**Simple Summary:**

Recently, there has been an emergence of a hypervirulent pathotype of *Aeromonas hydrophila,* vAh strain. The strain was responsible for the acute mass mortalities among catfish in the USA. One of the unique abilities of the vAh strain is to utilize *myo*-inositol as a sole carbon source and this ability has been linked to contribute to its virulence. Therefore, the present study was carried out to screen and assess the virulence of *myo*-inositol-utilizing strains among *Aeromonas* spp. in Malaysia. Out of the 124 *Aeromonas* isolates screened, only *A. dhakensis* strain 1P11S3 was able to utilize *myo*-inositol as a sole carbon source. The only *myo*-inositol-utilizing strain was compared with five non-*myo*-inositol-utilizing *Aeromonas* spp. in an experimental challenge test using red hybrid tilapia (*Oreochromis mossambicus* × *O. niloticus*). Our findings demonstrated that the most virulent strains were *A. dhakensis* strain 4PS2 and *A. hydrophila* strain 8TK3, followed by *A. dhakensis* strain 1P11S3 (the only *myo*-inositol-utilizing strain), *A. veronii* strain 6TS5, *A. caviae* strain 7X11 and the least virulent strain was *A. jandaei* strain 7KL3 under current disease model. Therefore, more data are needed to assess the influence of *myo*-inositol utilizing ability on the pathogenesis of *Aeromonas* spp.

**Abstract:**

The genus *Aeromonas* has been recognised as an important pathogenic species in aquaculture that causes motile *Aeromonas* septicaemia (MAS) or less severe, chronic infections. This study compares the pathogenicity of the different *Aeromonas* spp. that were previously isolated from freshwater fish with signs of MAS. A total of 124 isolates of *Aeromonas* spp. were initially screened for the ability to grow on M9 agar with *myo*-inositol as a sole carbon source, which is a discriminatory phenotype for the hypervirulent *A. hydrophila* (vAh) pathotype. Subsequently, LD_50_ of six selected *Aeromonas* spp_._ were determined by intraperitoneal injection of bacterial suspension containing 10^3^, 10^5^, and 10^7^ CFU/mL of the respective *Aeromonas* sp. to red hybrid tilapias. The kidneys, livers and spleens of infected moribund fish were examined for histopathological changes. The screening revealed that only *A. dhakensis* 1P11S3 was able to grow using *myo*-inositol as a sole carbon source, and no vAh strains were identified. The LD_50–240h_ of *A. dhakensis* 1P11S3 was 10^7^ CFU/mL, while the non-*myo*-inositol utilizing *A. dhakensis* 4PS2 and *A. hydrophila* 8TK3 was lower at 10^5^ CFU/mL. Similarly, tilapia challenged with the *myo*-inositol *A. dhakensis* 1P11S3 showed significantly (*p* < 0.05) less severe signs, gross and histopathological lesions, and a lower mortality rate than the non-*myo*-inositol *A. dhakensis* 4PS2 and *A. hydrophila* 8TK3. These findings suggested that *myo*-inositol utilizing *A. dhakensis* 1P11S3 was not a hypervirulent *Aeromonas* sp. under current experimental disease challenge conditions, and that diverse *Aeromonas* spp. are of concern in aquaculture farmed freshwater fish. Therefore, future study is warranted on genomic level to further elucidate the influence of *myo*-inositol utilizing ability on the pathogenesis of *Aeromonas* spp., since this ability correlates with hypervirulence in *A. hydrophila* strains.

## 1. Introduction

Aquaculture is one of the most important sectors that provides valuable sources of protein, aside from generating income for certain countries. FAO [1] reported a new height in global aquaculture production in 2018, with 114.5 million tonnes in live weight, generating USD 263.6 billion of farmgate sale value. Malaysia has been relying on the aquaculture sector as an important source of revenue, and FAO [2] also states that fish is an essential food in Southeast Asian nations, such as Malaysia, Indonesia, Myanmar, Thailand, Vietnam, Philippines, and Cambodia. In 2018, it was reported that the red hybrid tilapia (*Oreochrommis niloticus × O. mossambicus*) and black tilapia (*Oreochromis* spp.) accounted for 30.7% of total freshwater production in Malaysia, wherein red hybrid tilapia alone constituted about 97% of total tilapia production [3]. Tilapia farming is preferred because of their resilience in many environmental conditions, disease resistance, marketability, and easier-to-produce market-sized fish using a varied range of feed, ranging from natural organisms to humanmade pellets [4,5].

However, disease outbreak is one of the main problems which resulted in economic, environmental and social impacts on aquaculture. Fish farms are vulnerable to losses due to outbreaks of bacterial infections such as motile *Aeromonas* septicemia (MAS) [6]. This disease is caused by members of the genus *Aeromonas*, such as *Aeromonas hydrophila*, *A. veronii*, *A. dhakensis*, *A. jandaei*, *A. sobria* and *A. caviae* [7,8,9,10,11,12]. The disease was described as having two forms, the acute hemorrhagic septicemia and the chronic ulcerative syndrome [13,14,15]. The genus *Aeromonas* plays an important role in diseases of aquaculture, with *Aeromonas* spp. showing a ubiquitous distribution in aquatic habitats including freshwater, seawater, estuaries and even in chlorinated water [16,17]. The ability of *Aeromonas* spp. to thrive in a wide range of temperatures, from 4 °C to 42 °C, and tolerate up to 5.5 g/L NaCl and in a range of pH from 5 to 10 contributes to their widespread distribution [18,19]. *Aeromonas* spp. are generally opportunistic pathogens that are normal residents of the fish gut microbiota [20].

Recently, hypervirulent vAh strains were identified in diseased carp species in China and from diseased channel catfish from the USA, which were found to cause acute persistent MAS [21]. It was reported as a primary pathogen [22,23] among grass carp (*Ctenopharyngodon idella*) and channel catfish (*Ictalurus punctatus*) suffering from mass mortalities without concurrent infection [21,24,25]. Most of the affected fish were of market size [23,26], while another study reported the strain in juvenile fish [27].

The vAh strains have an apparently unique ability among *A. hydrophila* strains to utilize *myo*-inositol as a sole carbon source and Hossain et al. [26] further reported that *myo*-inositol utilizing gene cluster was part of epidemic associated region in vAh strains. Therefore, this characteristic has been used to screen vAh isolates among *A. hydrophila* [24,26,28]. Nevertheless, the only known vAh strains are among *A. hydrophila*, while *A. dhakensis*, *A. hydrophila*, *A. jandaei*, *A. veronii*, *A. sobria* and *A. caviae* were believed to be unable to utilize inositol [10,29,30,31]. However, with more recent data, there is a possibility that horizontal gene transfer may have happened between vAh and other notorious virulent *Aeromonas* spp., such as *A. dhakensis.* Therefore, this study screens for the *myo*-inositol utilizing ability and evaluate its influence on the pathogenicity of non- and *myo*-inositol utilizing *Aeromonas* spp. in red hybrid tilapia based on the results of LD_50_ and histopathological changes.

## 2. Materials and Methods

### 2.1. Collection of Aeromonas spp.

A total of 124 *Aeromonas* isolates that were previously obtained from fish with chronic ulcerative infection were used. The isolates were identified in previous experiment [32] as *A. dhakensis* (*n* = 53), *A. hydrophila* (*n* = 25), *A. veronii* (*n* = 27), *A. caviae* (*n* = 10), and *A. jandaei* (*n* = 9) using primers that identified the *gyrB* gene [33]. The isolates were streaked onto tryptic soy agar (TSA) (Oxoid, Hampshire, UK) and further sub-cultured onto *Aeromonas* selective agar (ASA) (Oxoid, Hampshire, UK) and incubated at 30 °C overnight.

### 2.2. Screening for Myo-Inositol Utilizing Strains and Blood Agar Characteristics

All isolates were screened for the ability to utilize *myo*-inositol using both agar and broth methods. The M9 agar (Difco, Sparks, MD, USA) with *myo*-inositol was prepared as previously described [34]. A total of 30 mL of sterile 10% *myo*-inositol and 2 mL of 1.0 M MgSO_4_ were added into the agar after it was cooled down to 55 °C. Observation for *myo*-inositol utilizing strains was completed after 24 h of incubation at 30 °C for both agar and broth methods. The bacteria that grew beyond 48 h were discarded. The isolates were also screened using TSA with 7% horse blood to check for the type of haemolysis.

### 2.3. Acclimatization of Red Hybrid Tilapia

A total of 380 juvenile red hybrid tilapias (*Oreochromis mossambicus × O. niloticus*) with mean ± standard deviation (SD) weight of 70.00 ± 6.81 g were obtained from a local commercial fish farm. The fish were acclimatized in a 500 L fibre-glass tank for 14 days with constant aeration. Approximately 30% of the water was changed while water quality parameters were monitored daily. The mean ± SD of water parameters, including pH (6.71 ± 0.14), temperature (25.51 °C ± 0.04 °C), salinity (0.25 ± 0.01 ppt) and total dissolved solid (342.5 ± 5.7 mg/L) were determined using a YSI meter (Yellow Springs Instrument, Inc., Yellow Springs, OH, USA). Throughout the experiment, the fish were fed two times a day ad libitum with commercial feed containing 32% crude protein. All procedures involving fish in this study were approved by the Institutional Animal Care and Use Committee, Universiti Putra Malaysia (AUP No: UPM/IACUC/AUP-R076/2019).

### 2.4. Preparation of Different Concentrations of Aeromonas spp.

Six isolates of *Aeromonas* spp. were selected for the pathogenicity study based on the number of virulence genes present in each isolate [32]. The selected isolates were the *myo*-inositol utilizing *A. dhakensis* 1P11S3 and the non-*myo*-inositol utilizing *A. dhakensis* 4PS2, *A. hydrophila* 8TK3, *A. veronii* 6TS5, *A. caviae* 7X11, and *A. jandaei* 7KL3. The five non-*myo*-inositol utilizing *Aeromonas* spp. had highest number of virulence genes within each species (Table A1) [32]. Briefly, five colonies on ASA were randomly picked and inoculated into 1 L of tryptic soy broth (TSB; Oxoid), and incubated for 18 h at 30 °C. Serial dilutions were performed up to tenfold and 1 mL of TSB from each dilution was plated on TSA in triplicate, and further incubated at 30 °C overnight. The colony forming units per millilitre (CFU/mL) were determined using standard plate count.

### 2.5. Pathogenicity Study

For pathogenicity test, there were six *Aeromonas* spp. and a control group involved. Therefore, a total of 380 juvenile red hybrid tilapias were randomly distributed into glass tanks of 20 L (18 challenge tanks + 1 control tank). The fish were divided into six groups (6 groups × 3 bacterial dilutions × 20 fish/tank) of bacterial isolates and 1 group of control (*n* = 20). The fish were anaesthetised using MS222 (Western Chemical Inc., Ferndale, WA, USA) at a concentration of 150 µg/L [35], before being inoculated intraperitoneally with 0.1 mL of the suspension containing respective *Aeromonas* sp. at 10^3^ CFU/mL, 10^5^ CFU/mL and 10^7^ CFU/mL. Control fish were similarly inoculated with sterile TSB. All fish were observed for clinical signs and mortalities for a period of 10 days. The LD_50–240h_ was calculated using Probit analysis with SPSS 25 [36].

### 2.6. Histopathological Assessment

All moribund fish (groups with dilution of 10^7^ CFU/mL) were immediately sampled and prior to histopathological evaluation, the fish were checked for lesions and haemorrhages. The kidney, liver and spleen samples were collected (*n* = 5 samples/group) and fixed in 10% neutral-buffered formalin for at least 24 h before routinely processed using paraffin-embedded technique, sectioned at 4 µm, stained with haematoxylin-eosin (H&E) and examined under a light microscope. The severity of the gross and histopathological changes in each of the internal organs were described and scored as follows; none: 0% affected (0), mild: <30% affected (1), moderate: 30–60% affected (2) and severe: >60% affected (3) [37].

### 2.7. Statistical Analysis

The mean of histopathological scorings in selected internal organs of the fish were compared using Kruskal–Wallis H test (SPSS, IBM Corporation, NY, USA). Before carrying out the Kruskal–Wallis H test, normality of the variance was evaluated using Shapiro–Wilk test, respectively. The results were considered significant at *p* < 0.05.

## 3. Results

### 3.1. Characterization of Aeromonas spp.

Out of the 124 *Aeromonas* isolates tested in this study, one isolate, the *A. dhakensis* 1P11S3 was found to utilize *myo*-inositol. The growth on M9 agar was observed as early as 12 h post-incubation and could clearly be seen after 24 h, while the M9 broth became turbid after 24 h of incubation with *A. dhakensis* 1P11S3 (Figure A1). All isolates (*n* = 124) produced complete haemolysis (β-haemolysis) on 7% blood agar (*n* = 124).

### 3.2. Clinicopathological Changes of Infected Fish

In general, all *Aeromonas*-infected fish showed various degrees of typical gross lesions and clinical signs of aeromoniasis. These symptoms included irregular breathing and lethargy, reduced feed consumption, displayed erratic movement with loss of balance, isolating and swimming to the surface of the water. The gross lesions included swollen and haemorrhagic site of injection (Figure 1A), scale loss that exposed the underlying skin at the caudal fin (Figure 1B) and necrosis of the fins (Figure 1B).

Post-mortem examinations revealed hepatomegaly (Figure 1C) in a majority of the infected fish, while multifocal splenic infarction (Figure 1D) was observed in fish infected with *A. hydrophila* 8TK3. Some infected fish had a haemorrhagic spleen and swollen gall bladder (Figure 1D). Overall, fish infected by *A. hydrophila* 8TK3 and *A. dhakensis* 4PS2 showed more severe gross lesions than other isolates. All control fish exhibited neither clinical sign nor gross lesion.

### 3.3. Lethal Dose of Aeromonas spp. in Red Hybrid Tilapia

The rates of mortality following infection by different isolates of *Aeromonas* were summarized in Table 1. Mortalities were observed as early as 24 h post-challenge in all infected groups, except for *A. jandaei* 7KL3 and lasted for 216 h. By 240 h, the cumulative mortality of fish challenged with 10^3^ CFU/mL of bacteria ranged between 0% and 20%, between 20% and 60% with 10^5^ CFU/mL and between 35% and 95% with 10^7^ CFU/mL. The two highest cumulative mortalities (95% and 90%) involved non-*myo*-inositol utilizing *A. dhakensis* 4PS2 (10^7^ CFU/mL) and *A. hydrophila* 8TK3 (10^7^ CFU/mL), which was higher than the *myo*-inositol utilizing *A. dhakensis* 1P11S3 (10^7^ CFU/mL) with 55% cumulative mortality.

The LD_50–240h_ for all *Aeromonas* spp. were presented in Table 1. The lowest LD_50–240h_ of 10^5^ CFU/mL was for *A. hydrophila* 8TK3 and *A. dhakensis* 4PS2, lower than the 10^7^ CFU/mL of *A. dhakensis* 1P11S3, *A. veronii* 6TS5 and *A. caviae* 7X11, while the highest LD_50–240h_ of 10^11^ CFU/mL was calculated for *A. jandaei* 7KL3.

### 3.4. Histopathological Analysis

The livers of all infected fish appeared congested with necrosis of hepatocytes. These were frequently accompanied by moderate to severe hydropic degeneration of the hepatocytes (Figure 2B,C). Individual necrosis of the hepatocytes was most severe in fish infected by *A. hydrophila* 8TK3 (*p* < 0.05), followed by *A. dhakensis* 4PS2, *A. dhakensis* 1P11S3, *A. veronii* 6TS5, *A. caviae* 7X11, and *A. jandaei* 7KL3. Infiltration of inflammatory cells, haemorrhages and hemosiderosis were also observed in the liver of all groups (Figure 2D). In addition, liver of fish infected with *A. hydrophila* 8TK3 showed hepatic infarction (Figure 2E).

Normal splenic architecture was observed in control group with presence of numerous erythrocytes in the red pulp and mildly scattered MMC (Figure 3A). On the other hand, the spleens of infected fish appeared congested involving multiple blood vessels with numerous MMC (Figure 3B), and infiltration of inflammatory cells associated with multifocal areas of splenic necrosis (Figure 3C). Different *Aeromonas* spp. were observed to produce different severity of lesions, ranging from mild to severe. Spleens of fish infected by *A. hydrophila* 8TK3 showed complete loss of splenic architecture (Figure 3D). Fish infected with *A. jandaei* 7KL3 showed white pulp hypoplasia, affecting small multifocal areas (Figure 3E). In general, fish infected with *A. hydrophila* 8TK3 showed most severe splenic lesion (*p* < 0.05), followed by *A. dhakensis* 4PS2, *A. dhakensis* 1P11S3, *A. veronii* 6TS5, *A. caviae* 7X11 and *A. jandaei* 7KL3.

The kidneys of control fish showed normal architecture with intact, healthy tubular epithelia (Figure 4A). However, all groups of infected fish showed varying degree of tubular epithelial and glomerular degeneration and necrosis (Figure 4B–D). Necrosis of tubular epithelial cells was seen as either desquamation from the basement membrane or karyolysis of the nucleus (Figure 4C). The latter was only observed in fish infected by *A. hyrophila* 8TK3, *A. dhakensis* 4PS2, and *A. caviae* 7X11. Fish infected by *A. jandaei* 7KL3 showed perivascular oedema, accompanied by necrotic tubular epithelium and infiltration of inflammatory cells (Figure 4D). These lesions were not seen in fish infected by other species of *Aeromonas*. Additionally, the tubular epithelial changes were more severe among these groups (*A. dhakensis* 4PS2 and *A. hydrophila* 8TK3), where complete lysis of tubular epithelial cells was frequently observed (Figure 4E). All fish infected with *Aeromonas* spp. showed desquamation (Figure 4F) and atrophied glomerulus. Cumulatively, histopathological changes in the kidney were most severe in fish infected with *A. dhakensis* 4PS2 and *A. hydrophila* 8TK3 (*p* < 0.05), followed by *A. jandaei* 7KL3, *A. dhakensis* 1P11S3, *A. veronii* 6TS5, and *A. caviae* 7X11.

The result of all histopathological scoring of liver, spleen and kidney was presented in Table 2. In summary, the most severe histopathological changes were observed in liver (*A. hydrophila* 8TK3), spleen (*A. hydrophila* 8TK3) and kidney (*A. hydrophila* 8TK3 and *A. dhakensis* 4PS2) of the infected fish. The *myo*-inositol utilizing *A. dhakensis* 1P11S3 caused mild to moderate histopathological changes in liver, spleen and kidney of the red hybrid tilapia.

## 4. Discussion

Recent reports on hypervirulent vAh strains have raised concern, particularly on the possibility of the spread of this hypervirulent pathotype in different countries and farmed fish species. This study identified only one strain, *A. dhakensis* 1P11S3, has the ability to use *myo*-inositol as a sole carbon source, which to our knowledge represents the first report of *A. dhakensis* that could utilize *myo*-inositol. None of the *A. hydrophila* isolates could utilize *myo*-inositol, suggesting that the vAh pathotype is not present or at least not common among farmed freshwater fish in Malaysia.

The subsequent challenge study produced typical signs and gross lesions of *Aeromonas* infection as reported by Soto-Rodriguez et al. [38] and Saharia et al. [15] in Nile tilapia (*O. niloticus*) and Indian major carp (*Labeo rohita*). There was increasing rate of mortality with increasing bacterial concentration, since more toxins would be released that could damage the internal organs of the fish [10]. Infected fish died as early as 24 h and as late as 216 h, highlighting the ability of *Aeromonas* spp. to cause acute-to-chronic infection in fish. The histopathological changes in the internal organs of red hybrid tilapia were also typical following *Aeromonas* spp. infection. In liver, changes were manifested as necrosis of liver parenchyma [15,39,40]. Splenic necrosis was also observed, similarly reported by previous studies [39,40]. Necrosis of kidney was characterized by swollen, desquamated and vacuolated tubular epithelial cells with some cells having complete necrosis, while atrophied glomerulus were also reported in previous studies [15,40,41]. Other histopathological changes such as the presence of polymorphonuclear cells in the inter tubular area, along with proliferated cell of glomerulus, were not observed in this study [15,39]. Alternatively, the presence of haemocyte aggregation and congestion in all selected internal organs indicated that liver, spleen and kidney are the target organ for *Aeromonas* systemic infection [16,39,40]. In addition, histopathological changes in the liver, spleen, and kidney were highly suggestive of acute to subacute cellular injury. It is worth mentioning that no abnormalities were observed following infection by the *myo*-inositol utilizing *A. dhakensis* 1P11S3, indicating that inositol catabolism per se did not contribute to *A. dhakensis* pathogenesis in tilapia with current disease model. Furthermore, all isolates used in this study were obtained from diseased fish, without any relation with epidemic outbreak [32]. On the other hand, *myo*-inositol utilizing *A. hydrophila* vAh, that were isolated during epidemic outbreak among catfish, have been reported to cause severe clinical MAS symptoms with rapid onset of mortality [27]. Zhang et al. [42] reported 90% mortality among catfish challenged with vAh strains; however, the disease model included clipping of adipose fin and immersion in a high dose of 10^7^ CFU/mL vAh, processes that were not conducted in the present study. Therefore, modification of the challenge model should be conducted in future study.

Based on the LD_50–240h_, the most virulent *Aeromonas* strains were *A. dhakensis* 4PS2 and *A. hydrophila* 8TK3 (10^5^ CFU/mL), while the least virulent was *A. jandaei* 7KL3 (10^11^ CFU/mL). Previous reports revealed the LD_50_ of *A. dhakensis* was between 10^2^ and 10^5^ CFU/mL when challenged in zebrafish (*Danio rerio*), rainbow trout (*Oncorhynchus mykiss*) and pacu (*Piaractus mesopotamicus*) [30,43,44]. Similarly, few studies reported LD_50_ < 10^6^ CFU/mL for *A. hydrophila* in zebrafish, rainbow trout and tilapia [45,46,47].

When the LD_50_ of each isolate was compared with the presence of virulence genes [32], *A. dhakensis* 4PS2 and *A. hydrophila* 8TK3 were found to have all of the virulence genes that were tested. In addition, both strains also possessed the *aer* gene, which might play a significant role in the virulence of *Aeromonas.* On the other hand, *A. dhakensis* 1P11S3 (*myo*-inositol utilizing *Aeromonas*) and *A. caviae* 7X11 shared the same number of virulence genes (only 62.5% of the tested virulence genes), whereas *A. veronii* 6TS5 had fewer virulence genes (50% of the tested genes), while sharing the same LD_50–240h._ In addition, those three strains also harboured the *fla* and *ela* genes, while *A. jandaei* harboured the fewest virulence genes (37.5% of the tested genes) and was found to lack the *ahp*, *fla*, *alt*, *lip* and *aer* genes. While there are many factors that can contribute to pathogenesis beyond the presence or absence of genes (e.g., gene expression), it was noteworthy that the presence of the *aer* gene was the most highly correlated with *Aeromonas* strain virulence. In addition, Zhang et al. [48] reported that aerolysin as one of the most potent toxins produced by *A. hydrophila,* where acute mortality of catfish was observed as early as 3 h following exposure to either proaerolysin (inactive form) or aerolysin (activated form).

Following the challenge test, *A. dhakensis* 1P11S3 (*myo*-inositol utilizing) appeared to lack significant virulence under current tested conditions; however, it is possible that the ability to utilize *myo*-inositol may contribute to *A. dhakensis* persistence within aquaculture pond ecosystems through the use of *myo*-inositol, which is present in plant-based feeds that contain phytate [49]. Therefore, to arrive at the conclusion that *myo*-inositol utilizing ability did not translate to increased virulence in *Aeromonas* spp., more research and verification is required. We postulated that there may be different genomic representation of *myo*-inositol utilizing vAh with our *A. dhakensis* 1P11S3, since both are of different *Aeromonas* spp. In vAh strains, *myo*-inositol metabolism was found to be isolated, instead of dispersed like *Corynebacterium glutamicum* and *Caulobacter crescentus* [50,51]; the cluster was found in genomic island 2, an epidemic-associated region. The genomic island 2 was also determined to encode other putative virulence factors such as gene *bpIA* (probable oxidoreductase) and *hitC* (iron (III) ABC transporter) [27]. In addition, our study elucidated the influence of eight virulence genes on the virulence of *Aeromonas* sp. based on the result of the challenge test. However, the possibility of other virulence factors that might have even greater significance in *myo*-inositol utilizing *Aeromonas* sp. should not be neglected. Rasmussen-Ivey et al. [21] reported that more virulent vAh isolates lacked several components of type VI secretion system (T6SS). Meanwhile, T6SS in *A. dhakensis* was very crucial for its virulence [52], but in *A. hydrophila,* it was determined to have a least pronounced effect [53]. Nevertheless, horizontal gene transfer may have occurred between vAh strain and *A. dhakensis* 1P11S3, which to the author’s knowledge, no study has reported this ability among *A. dhakensis*. Therefore, future studies should be conducted immediately on the whole-genome sequencing of *A. dhakensis* 1P11S3, which can potentially reveal the in-depth state of the virulence of the isolate, alongside the genotypic properties of *myo*-inositol catabolism cluster of the isolate. The study also indicates that there are diverse, pathogenic *Aeromonas* spp. that contribute to disease prevalence within farmed freshwater fish in Malaysia.

## 5. Conclusions

In conclusion, this study reports for the first time that *A. dhakensis* could utilize *myo*-inositol as a sole carbon source. A subsequent pathogenicity study revealed that the non-*myo*-inositol utilizing *A. hydrophila* and *A. dhakensis* was the most pathogenic among the tested isolates. While no vAh strains were identified in this study, other virulent *Aeromonas* spp. isolates that were highly virulent in farmed fish were identified.

## Figures and Tables

**Figure 1 biology-10-01192-f001:**
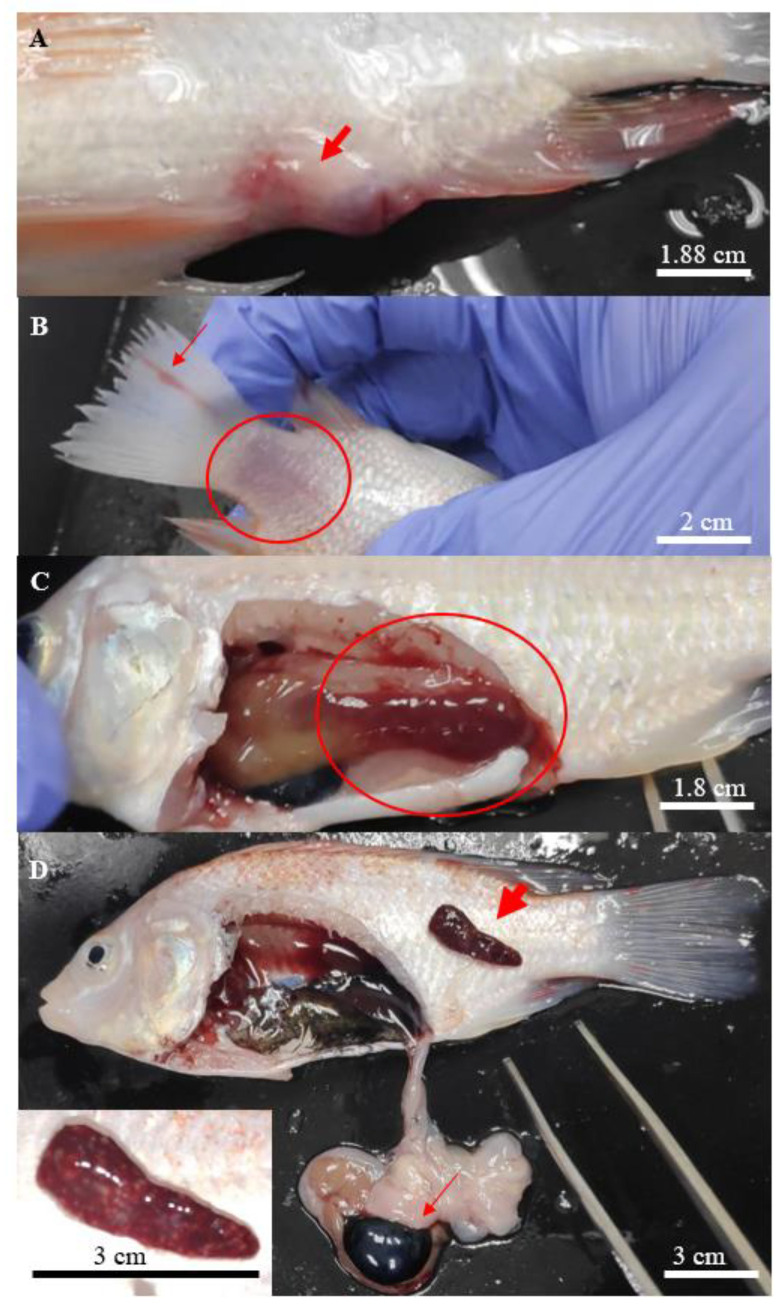
Clinical signs found on the hybrid red tilapias infected with *Aeromonas*. (**A**) focal haemorrhage on the site of injection, (**B**) scale loss (circle) at the caudal peduncle with rotting caudal fin (thin arrow), (**C**) liver with deep haemorrhage (circle), and (**D**) haemorrhage with multiple white spots observed on the enlarged spleen (thick arrow), enlarged gallbladder (thin arrow), with a close-up picture of the affected spleen at the bottom left.

**Figure 2 biology-10-01192-f002:**
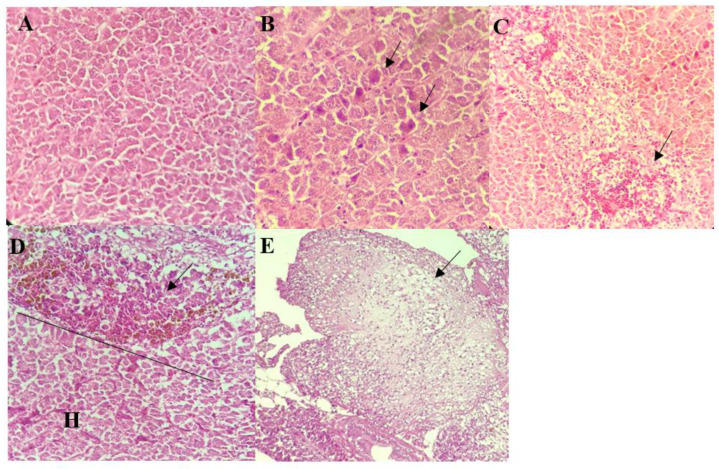
Histological findings of the liver of control and infected red hybrid tilapias by *Aeromonas* spp. (**A**) Liver of control fish showed normal liver architecture and hepatocytes, 400×. (**B**) Individual hepatocyte necrosis exhibiting intense cytoplasmic eosinophilia (arrows), 400×. (**C**) Extensive hydropic degeneration and necrosis of cells surrounding a central vein (arrow), 400×. (**D**) Infiltration of inflammatory cells. Note that some inflammatory cells are showing brownish cytoplasmic pigmentation suggestive of hemosiderosis (arrow) ((H): healthy cells), 400×. (**E**) Hepatic infarction showing complete loss of liver architecture (arrow), 400×.

**Figure 3 biology-10-01192-f003:**
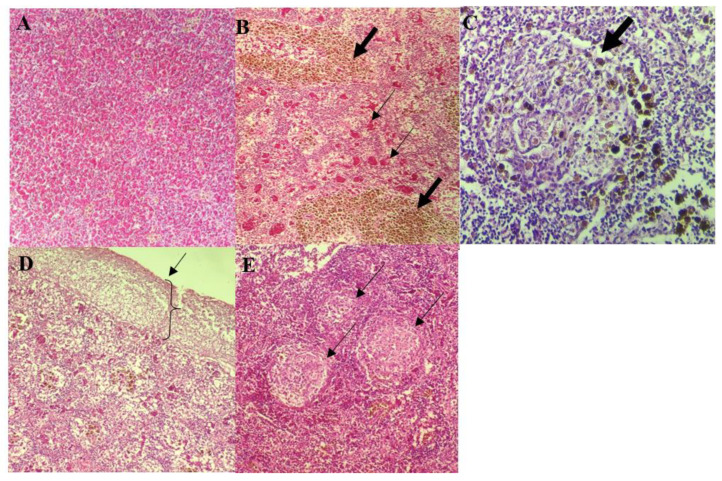
Histological findings of the spleen of control and infected red hybrid tilapias by *Aeromonas* spp. (**A**) Normal spleen of control fish showing normal splenic architecture, 200×. (**B**) Presence of abundant MMC (thick arrow) and multiple areas of splenic vascular congestion (thin arrow), 400×. (**C**) Severe, focal area of necrosis (arrow), with the presence of MMC, 400×. (**D**) Severe necrosis observed mainly at the splenic surface. 200×. (**E**) Multifocal areas of splenic white pulp hypoplasia (arrow), 200×.

**Figure 4 biology-10-01192-f004:**
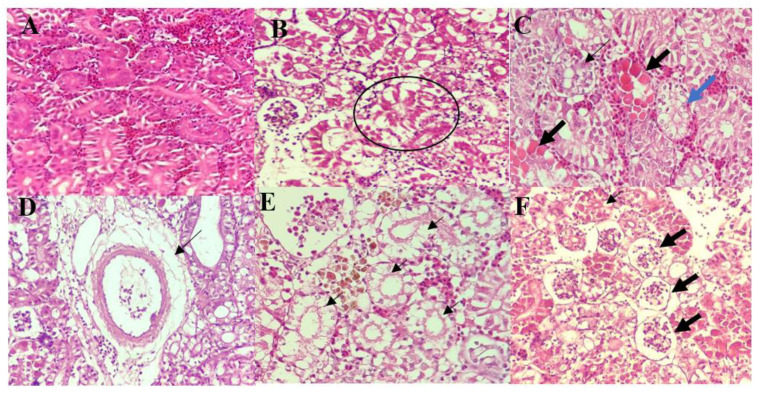
Histological findings of the kidney of control and infected red hybrid tilapias by *Aeromonas* spp. (**A**) Kidney of control fish showing normal and intact renal epithelial cells, 400×. (**B**) Hydropic degeneration (circle) of tubular epithelium, 400×. (**C**) Karyolysis and severe cytoplasmic eosinophilia of the tubular epithelial cells (thick arrow), with detachment of tubular epithelium from the basement membrane (thin arrow), and severe vacuolation (blue arrow), 400×. (**D**) Dilution of tissue surrounding a blood vessel suggestive of perivascular oedema, 400×. (**E**) Severe lysis of tubular epithelium (thin arrow), 400×. (**F**) Necrosis of glomerulus (thick arrow) and karyolysis of tubular epithelial cells (thin arrow), 400×.

**Table 1 biology-10-01192-t001:** The number of mortalities of red hybrid tilapias infected with six strains of *Aeromonas* spp.

Species	Bacterial Concentration	Number of Deaths at Specific Time (Hour Post Infection)										Sample Number	Cumulative Mortality (%)	LD_50–240h_ (CFU/mL)
	(CFU/mL)	24	48	72	96	120	144	168	192	216	240			
*A. dhakensis* 1P11S3 *	10^3^	2	0	0	0	0	0	0	0	0	0	20	2 (10)	1 × 10^7^
	10^5^	2	1	1	0	0	0	0	0	0	0	20	4 (20)	
	10^7^	2	4	1	4	0	0	0	0	0	0	20	11 (55)	
*A. dhakensis* 4PS2	10^3^	0	0	0	1	0	0	0	0	0	0	20	1 (5)	1 × 10^5^
	10^5^	1	5	6	8	0	0	0	0	0	0	20	8 (40)	
	10^7^	11	0	3	4	1	0	0	0	0	0	20	19 (95)	
*A. hydrophila* 8TK3	10^3^	0	3	0	0	0	0	0	0	0	0	20	3 (15)	1 × 10^5^
	10^5^	0	2	6	2	1	0	1	0	0	0	20	12 (60)	
	10^7^	7	2	6	1	1	0	1	0	0	0	20	18 (90)	
*A. veronii* 6TS5	10^3^	0	1	0	0	1	0	0	0	0	0	20	2 (10)	1 × 10^7^
	10^5^	2	3	1	0	0	0	0	0	0	0	20	6 (30)	
	10^7^	3	0	1	3	0	0	1	0	1	0	20	9 (45)	
*A. caviae* 7X11	10^3^	0	0	0	0	0	0	0	0	0	0	20	0 (0)	1 × 10^7^
	10^5^	2	0	0	1	0	2	0	1	0	0	20	6 (30)	
	10^7^	1	3	3	1	1	1	0	0	0	0	20	10 (50)	
*A. jandaei* 7KL3	10^3^	0	2	0	1	1	0	0	0	0	0	20	4 (20)	1 × 10^11^
	10^5^	0	3	0	0	0	2	0	0	0	0	20	5 (25)	
	10^7^	0	3	2	0	0	0	1	1	0	0	20	7 (35)	
Control (sterile TSB)	0	0	0	0	0	0	0	0	0	0	0	20	0 (0)	

* Indicate isolate that can utilize myo-inositol as a sole carbon source.

**Table 2 biology-10-01192-t002:** Summary of histopathological changes scoring in liver, spleen and kidney of red hybrid tilapias following experimental infection by *Aeromonas* spp.

Species	Organ		
	Liver	Spleen	Kidney
*A. dhakensis* 1P11S3 (10^7^ CFU/mL) *	1.0 ± 0.0 ^a^	2.0 ± 0.1 ^a^	1.0 ± 0.0 ^a^
*A. dhakensis* 4PS2 (10^7^ CFU/mL)	2.0 ± 0.1 ^c^	2.0 ± 0.0 ^a^	3.0 ± 0.0 ^c^
*A. hydrophila* 8TK3 (10^7^ CFU/mL)	3.0 ± 0.0 ^b^	3.0 ± 0.0 ^b^	3.0 ± 0.0 ^b^
*A. veronii* 6TS5 (10^7^ CFU/mL)	1.0 ± 0.0 ^a^	2.0 ± 0.0 ^a^	1.0 ± 0.1 ^a^
*A. caviae* 7X11 (10^7^ CFU/mL)	1.0 ± 0.0 ^a^	2.0 ± 0.0 ^a^	1.0 ± 0.0 ^a^
*A. jandaei* 7KL3 (10^7^ CFU/mL)	1.0 ± 0.0 ^a^	1.1 ± 0.1 ^c^	2.1 ± 0.1 ^d^
Control (sterile TSB)	0.0 ± 0.0 ^d^	0.0 ± 0.0 ^d^	0.0 ± 0.0 ^e^

Data are shown as mean ± SE. 0 = normal; 1 = mild; 2 = moderate; 3 = severe, * = indicate isolate that can utilize *myo*-inositol as a sole carbon source. The number of samples was *n* = 5 for each group. ^a,b,c,d,e^ Different letters represent significant differences (*p* < 0.05) within the same column only.

## Data Availability

Not applicable.

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
