# Peer review of "Comparative Pathogenicity of Aeromonas spp. in Cultured Red Hybrid Tilapia (Oreochromis niloticus × O. mossambicus)"

_biology, 2021, doi:10.3390/biology10111192_

Round 1

Reviewer 1 Report

Comparative pathogenicity of Aeromonas spp. in cultured red hybrid tilapia (Oreochromis niloticus × O. mossambicus) is dealing with a serious problem of aquaculture. Even though authors bring new data, the contribution of results is based on my humble opinion insufficient for a journal with high IF such as Biology.

Minor comments:

-number that represents affiliations should be placed upper

  • abbreviation "US" should be replaced with "USA"
  • the first sentence of Simple summary needs to be rewritten add a comma after vAh strain and start a new sentence with "The strain...
  • " Therefore, THE present study ..."
  • Introduction can be longer, add why you chose the red hybrid tilapia as a model organism, also the first two sentences from the discussion can be moved to the introduction.
  • Technical mistake in section 5. Conclusion, you have two commas.

Reviewer 2 Report

General comment

The manuscript biology-1323133 entitled “Comparative pathogenicity of Aeromonas spp. in cultured red hybrid tilapia (Oreochromis niloticus × O. mossambicus)” aimed to compare the pathogenicity of the different Aeromonas spp. that were previous isolated from freshwater fish with signs of MAS. A total of 124 isolates of Aeromonas spp. were initially screened for the ability to grow on M9 agar with myo-inositol as a sole carbon source, which is a discriminatory phenotype for the hypervirulent A. hydrophila (vAh) pathotype. Subsequently, LD50 of six selected Aeromonas spp. were determined by intraperitoneal injection of bacterial suspension of Aeromonas sp. to red hybrid tilapias. The main organs of infected moribund fish were examined for histopathological changes. The study is interesting and fits well with the aims of the journal. However, there are some issues that need to be addressed before it can be accepted for publication in Biology.

Major comments

  • Introduction is too short. the introduction should place the study in a broad context and highlight why it is important. It should define the purpose of the work and its significance, including specific hypotheses being tested. In particular, the Authors have to better specify the objectives of the study.
  • Line 102. Did you perform a health monitoring (bacteriological, virological and parasitological exams) on fish before the experiment? Please, specify. This is crucial to perform an experiment involving fish.

Minor comments

  • Line 2: Aermonas in italics
  • Affiliation numbers: superscript
  • Line 102: SD (standard deviation)
  • Line 134. Add a statement that you performed an anatomopathological examination on fish
  • Line 145. You used a non-parametric test since your data were not normally distributed. Please, add a statement that you performed a statistical test to check the normality.
  • Line 153. Remove “.” before “Clinicopathological”
  • Line 220. Figure 1, add the bar scales

Reviewer 3 Report

Dear authors

Your manuscript about Aeromonas pathogenicity shows differences in virulences between the used species. Overall it is well written although there are some revisions to address.

Major revisions:

The introduction should have some information about red hybrid tilapia, the species needs to be introduced and explained why authors choose this fish

The material and methods needs some clarification: fish are divided into 6 groups of bacterial isolates and each group has 3 dilutions, so were there 18 tanks + 1 for control? Samples were collected 5/group, here it is not clear from which dilution (although in table 2 in the results it shows only the highest concentration). Were fish fed during the 10 days of the pathogenicity study?

Minor changes:

Line 23-24: in the US

Line 35: has been recognised as an important pathogenic species in aquaculture

Line 48: and a lower mortality rate

Line 63: please write MAS full out at first (although this has been done in the abstract)

Line 102: how many fish were used in the acclimatisation step (380? as explained later)

Line 105: water was changed

Line 113: for the pathogenicity

Line 128-129: at a concentration of 150 ug/L, before being inoculated intraperitoneally

Line 289: please change was to were

Line 423: were also typical

Line 427: studies

Line 453: role in the virulence

Line 455: please delete at

line 473: the genomic island is a epidemic-associated region

Line 491: for the first time that A.dhakensis could utilize

Line 493: were the most

Reviewer 4 Report

This manuscript reports results from laboratory infections of cultured red tilapia with an array of different motile Aeromonas isolates from diseased fish from Malaysian aquaculture. The authors tried to test the hypothesis that the ability of Aeromonas isolates to use myo-inositol as sole carbon source is linked to the virulence of these isolates to fish. Among their bacterial isolates, one A. dharkensis strain was able to utilise myo-inositol. In the subsequent infection experiment, this isolate was of moderate virulence compared to a further A. dharkensis strain and an A. hydrophila strain.

The manuscript contributes to the discussion on factors, which might determine virulence in motile aeromonads for fish and is of value, because it adds results of an infection experiment to results of biochemical and molecular biological analyses of respective bacterial strains. Overall, the manuscript is well written and presents a valid array of data. The results of the infection experiment are well documented and discussed. I have just a few minor comments:

Statistical evaluation of the histopathological changes in tissues of infected tilapia: In materials and methods, page 4, line 144, it is said that the results were compared using a Mann-Whitney U-Test. This test allows to compare data from two groups, however here findings from several groups were compared. For this an ANOVA on ranks should be performed.

Discussion, line 450 ff: discussion of the presence of virulence genes in the isolates tested in the infection experiment: among the currently tested isolates, bacteria, which possessed the aer gene, were highly virulent to infected tilapia. This corresponds to findings of Zhang and co-workers (2013, Vet Mic 165 (3-4), 478-482), which describe a pre-aerolysin from A. hydrophila and demonstrate a lethal activity of the recombinantly expressed protein encoded by this gene to catfish. This article adds an important aspect to the current study and should be considered in the discussion.

Round 2

Reviewer 1 Report

The authors improved the paper according to my suggestions, however, the paper was not substantially changed, therefore the final decision is on handling the Editor.  

Reviewer 2 Report

The Authors have addressed all my comments. The MS is now ready for publication in Biology.

Reviewer 3 Report

Dear authors

Your manuscript has improved, showing different pathogenicity of Aeromonas on tilapia, which might be of interest to aquaculture farmers

please change the following minor comments

Line 60: on the aquaculture

Line 70: which results in